# Seasonal variation in fish school spatial distribution and abundance under the Kuroshio regular pattern and the large meander in Suzu coastal waters

Yanhui Zhu[1]*, Kenji Minami[2], Yuka Iwahara[3], Kentaro Oda[3], Koichi Hidaka[3], Osamu Hoson[3], Koji Morishita[3], Masahito Hirota[3], Sentaro Tsuru[4], Hokuto Shirakawa[5], Kazushi Miyashita[6]

1 Graduate School of Environmental Science, Hokkaido University, Hakodate, Hokkaido, Japan, 2 Estuary Research Center, Shimane University, Matsue, Shimane, Japan, 3 Marine Fisheries Research and Development Center (JAMARC), National Research and Development Agency, Japan Fisheries Research and Education Agency, Yokohama, Kanagawa, Japan, 4 Yamaguchi Regional Alliance office, Planning and Coordination Department, Fisheries Technology Institute, National Research and Development Agency, Japan Fisheries Research and Education Agency, Shimonoseki, Yamaguchi, Japan, 5 Fisheries Stock Assessment Center, Fisheries Resources Institute, National Research and Development Agency, Japan Fisheries Research and Education Agency, Niigata, Japan, 6 Field Science Center for Northern Biosphere, Hokkaido University, Hakodate, Hokkaido, Japan

* zhuyanhui0817@eis.hokudai.ac.jp

**Data Availability Statement:** All relevant data are within the paper and its Supporting Information files.

## Abstract

The Kuroshio Current can take two paths; usually it follows the regular pattern but occasionally it follows a pattern known as the large meander. In this study, we investigated the abundance of fish that migrate to coastal waters and the spatial distribution of fish schools under both Kuroshio patterns in Suzu district, Kochi prefecture, where the set net is the main fishery industry. We clarified the seasonal variation in the density and distribution of fish schools using a quantitative echo sounder. The effects of the Kuroshio large meander (LM) depended on the season. There was no effect of current pattern in summer or autumn, but in winter and spring the LM altered the marine environment and fish distributions. Cold water masses were formed in the survey area during winter and spring during the LM, and the water temperature dropped significantly compared with during the Kuroshio non-large meander (NLM). This altered the fish species and the distribution of fish schools in the survey area. The catches of Japanese horse mackerels (*Trachurus japonicus*) and Yellowtails (*Seriola quinqueradiata*) were much higher during the LM compared with those during the NLM. Unlike these two species, the small-sized pelagic fishes in spring has decreased significantly during the LM.

## Introduction

The Kuroshio Current is one of the warmest water currents in the world, flowing off the southern part of Shikoku, which is one of the main Japanese islands. Abundant heat and larvae/

**Funding:** Marine Fisheries Research and
Development Center, Japan Fisheries Research and
Education Agency (Empirical Research Project for
Marine Fisheries Resource Development: Set-net in
Suzu, Kochi prefecture, Fiscal Year 2016-2018)
http://jamarc.fra.affrc.go.jp/index-e.html Funders:
Decision to publish The funders had role in study
design, data collection and decision to publish.

**Competing interests:** The authors have declared
that no competing interests exist.

juvenile of fish are transported to the Shikoku sea area by the Kuroshio Current [1]. Additionally, due to the turbulence of the Kuroshio Current, the abundance of nutrients contained in the deep sea are rolled up to the shallow layer, resulting in sustained primary production in the Shikoku sea area [2]. Thus, the Shikoku sea area is rich in plankton and is a suitable spawning ground and habitat for fish, making it a good fishing ground for the food chain [3]. The highly productive Kuroshio ecosystem supports the fisheries and livelihood in Shikoku.

The Kuroshio Current has a wide variety of flow patterns (Fig 1), which can be primarily classified into two groups: 1) non-large meander (NLM) along the coast of Southern Japan, and 2) large meander (LM) curve to the south off the coast of the Kii Peninsula [4–7]. The LM appears once in an approximately ten-year cycle [8], and continues for more than a year [9]. The latest LM occurred in August 2017 for the first time in 12 years [10], and the period is the second longest in history [9]. Since cold water masses and warm water masses are formed along with LM [8, 9], the marine environment and species composition have changed significantly [11, 12], greatly affecting the Kuroshio ecosystem [13, 14]. However, when the LM occurs, the target space for investigation is extremely extensive to sufficiently assess the influence on the fishing grounds, unlike during years in NLM [15]. Therefore, clarifying the ecological changes that occur during the LM is indispensable for fisheries in Shikoku, part of Kuroshio basin.

Tosa Bay, situated in the southern area of Shikoku, is one of the sea areas that is greatly affected by fluctuations in the Kuroshio Current [16]. The fishing industry in Tosa Bay is mainly set net, towed net, and fishing fisheries. In particular, the set net fisheries have developed and occupy the top position among the coastal fisheries [17, 18]. Most of the set net catches in Tosa Bay are small pelagic fish such as sardines, horse mackerels and mackerels, but highly migratory fish such as Yellowtail (*Seriola quinqueradiata*) and bonitos are also caught [18]. In addition, the fish composition structure in Tosa Bay is seasonal [19], and the seasonal changes tend to differ depending on the yearly berthing and undocking of the Kuroshio Current [20]. Set net fisheries is a passive net fishing technique in which the catch is mainly dependent on the density and distribution of fish schools migrating into the sea area where the set net is installed [21]. Therefore, seasonal changes and spatial evaluation in biological information as the Kuroshio Current changes is necessary in set net fisheries.

For understanding the density and three-dimensional distribution of fish schools in seawater, acoustic technology is commonly used. It enables surveys of fisheries resources over a wide area to be easily conducted in a short period [22, 23]. A quantitative echo sounder can be used to convert the acoustic parameters of the density and positional distribution of fish schools into their quantified values [24]. Therefore, it is commonly used in aquatic resource surveys as a scientific assessment method. In recent years, walleye pollack (*Gadus chalcogrammus*) in Pacific waters [25], Japanese pearl sides (*Maurolicus japonicus*) around Oki Islands [26], and other fish have been visualized and quantified using a quantitative echo sounder.

Therefore, we aimed to assess the seasonal variation in fish abundance and species that migrated into the coastal area of Tosa Bay during the LM, and seasonal variation in the spatial distribution of fish schools. We compared the density and spatial distribution of fish schools, and composition of fish species during the same season in a year when NLM occurred and during a year when LM occurred. The density and spatial distribution of fish schools in the target areas were measured using a quantitative echo sounder. Data on the catch and marine environment were then acquired for examining correlations between the marine environment and biological information. Additionally, based on the acquired information, we examined the influence of the LM on the set net fisheries.

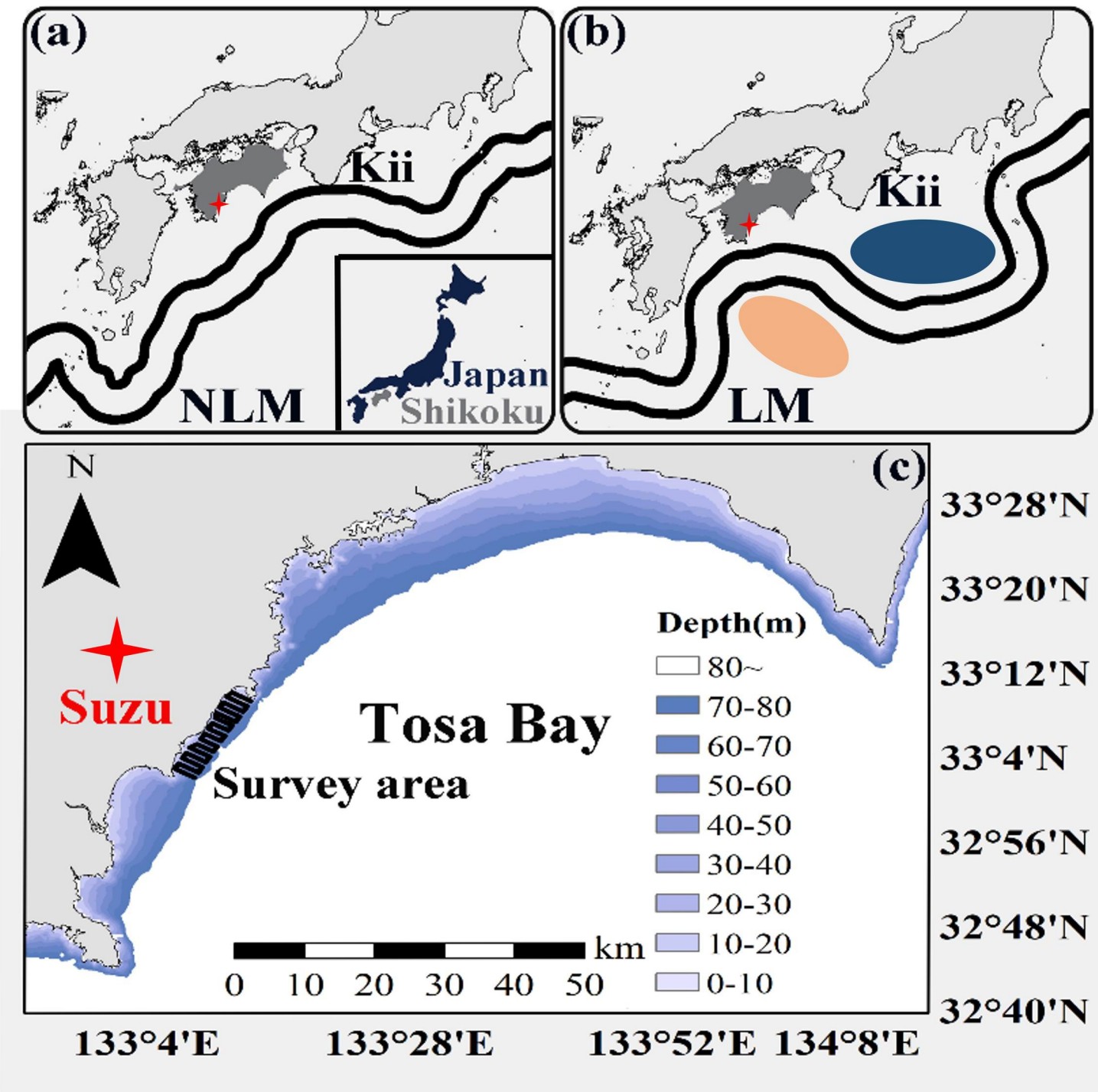

**Fig 1.** (a), (b) Two typical paths of the Kuroshio south of Japan: the offshore non-large meander (NLM) path on February 8, 2017, and the large meander (LM) path on February 25, 2018; based on the data from Hydrographic and Oceanographic Department Japan Coast Guard (JCG: https://www1.kaiho.mlit.go.jp/jhd.html). Blue cold water mass and pink warm water mass were modified from Miyama (2019) [9]. (c) The study area and survey line. The solid line shows the hydro acoustic transects, with a total length of 64 km. The triangle marks the location of the set net.

## Methods

The coastal waters of Suzu district in the western area of Tosa Bay was studied (Fig 1), where a set net was installed. A survey was conducted once every summer (August), autumn (November/December), winter (February), and spring (May) during the period from August 2016 to May 2018. Since the LM occurred in August 2017, eight surveys were used to understand the bio-phenology of the organisms migrating before and after the appearance of the LM. The period from August 2016 to May 2017 was defined as the NLM year, and the period from August 2017 to May 2018 was defined as the LM year. Each survey was conducted over 2 to 3 days, and conducted during daytime hours when pelagic species of fish, which are the main targets of set net fishing, tend to form fish schools [27]. The survey area was a relatively wide region of 5 km ×25 km and the water depth was 10 m—70 m.

The acoustic survey was conducted using a quantitative echo sounder with a frequency of 38 kHz (KSE300, SONIC) (Table 1), installed on the starboard side of a set net fishing ship (Suzu-Maru, 9.91 tons) along the survey line set in the target area (Fig 1). The quantitative echo sounder was calibrated before the start of each survey. When the hull of the ship comes into contact with the sea surface, the bubbles created can interfere with the transmission and reception of acoustic waves [28]. To avoid this interference, acoustic information was collected with a transducer installed at a point 1 m below the water surface at a vessel speed of 5 knots—7 knots. Positional information was recorded simultaneously with acoustic information using a differential global positioning system GPS (Tempest, Trimble). To understand the appearance frequency of fish species inhabiting the target area or migrating into these areas, the composition and ratios of fish species caught during each survey period and in each season were examined based on set net catch data (excluding summer season due to typhoons). In addition to the set net catch data, organisms were collected by fishing approximately 10 min after the echo sounder identified a fish school for determining the composition of fish species for each fish school. To grasp the marine environment around the set net, the temperature and salinity were then measured for one day during each survey period in the target areas using a RINKO-Profiler (ASTD102, JFE Advantech). The temperature and salinity from the sea surface to the seabed at the measuring points was acquired.

Echoview ver. 7.1 (Sonardata Tasmania Pty Ltd.) was used to analyze the acoustic data. Only the fish schools were analyzed (Fig 2). Area backscattering intensity ($Sa$: dB) and volume backscattering intensity ($Sv$: dB) were used as indicators of fish school density [29]. For eliminating the influence of noise generated by microorganisms such as plankton and suspended matter in seawater, the threshold for analysis was set at—60 dB [30]; any value lower was excluded from the results. Spatial information integrated with the geographic information system (GIS) of fish school, was then visualized [23].

To estimate the density of organisms migrating in the target areas, the mean $Sa$ linear value of all the target areas and the compositions of fish schools caught by the set net during the

**Table 1. Setting for the quantitative echo sounder (KSE300) with 38 kHz transducer in all surveys.**

| Variable | Specification |
|---|---|
| Transducer | T-178 |
| Frequency(kHz) | 38 |
| Beam type | Split |
| Beam width(°) | 8.5 |
| Pulse width(ms) | 0.6 |
| Ping rate(s) | 0.2 |

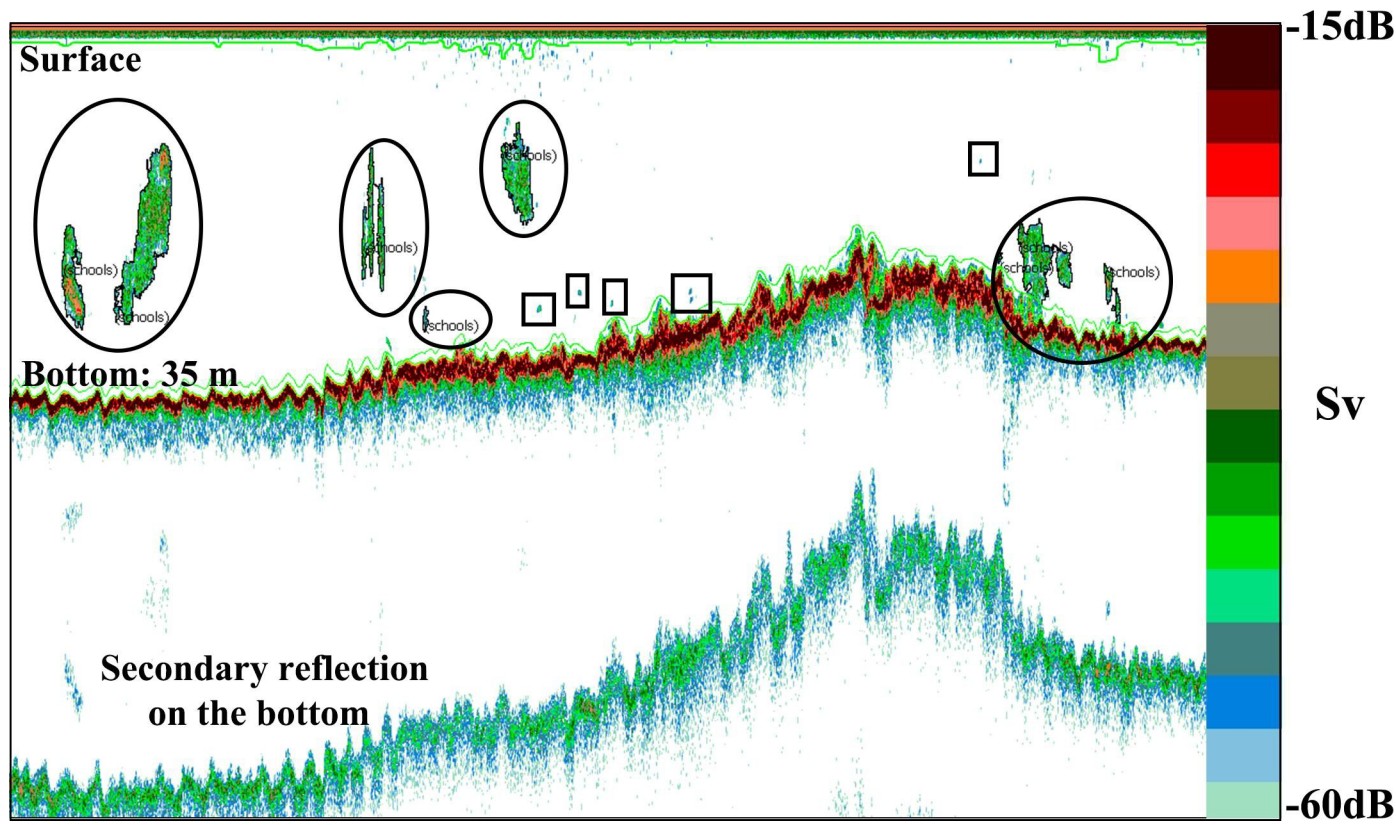

**Fig 2. The echogram shows object fish schools and non-object fish schools.** The circled area indicates the object fish school, and the square area indicates non—object fish school.

survey period were used [31]. Assuming that the compositions of fish schools caught by the set net reflect the compositions of fish species in the target areas, the fish school density in the target areas was calculated using formulas (1), (2), (3), and (4).

$$\sum n_k = \frac{Mean(sa)}{\sum ts_k * P_k} \tag{1}$$

$$N = S_{area} * \sum n_k \tag{2}$$

$$TS_k = 10 * \log(ts_k) \tag{3}$$

$$Sa = 10 * \log(sa) \tag{4}$$

In these equations, $n_k$, $ts_k$, and $p_k$ are the mean density / unit area, liner value of target strength / kg, and ratio to the total catch of fish species k, respectively, and $N$ and $S_{area}$ are the estimated fish school density and the size of the target areas, respectively. Fish school densities were calculated using the target strength / kg, $TS_k$, which was—39.3 dB for Japanese horse-mackerel (*Trachurus japonicus*), - 30.8 dB for chub mackerel (*Scomber japonicus*), - 32.9 dB for Yellowtail (*Seriola quinqueradiata*), - 34.6 dB for Japanese sardine (*Sardinops melanostictus*), and—31 dB for species that have not been recorded [31]. A comparison between the estimated fish school densities and the set net catch data was then carried out to validate the acoustic data.

To determine fish school distribution patterns, non-parametric generalized additive models (GAMs) with a smooth spline function were used [32]. GAMs are not restricted by linear relationships, and are flexible regarding the statistical distribution of the data [33]. In this study, positional information was added to the acoustic information for estimating the structure of distributed fish schools using GAMs, with the function *gam* in the package 'mgcv' in R ver. 3.5.0 (R Core Team 2017). In the horizontal structure, the response variable is the mean *Sa* value with horizontal intervals of 10 m, and the explanatory variable is the distance from the coast to *Sa* cell (*L*, m). In the vertical structure, the response variable is the mean *Sv* values with horizontal and vertical intervals of 10 m, and the explanatory variable is the median depth of *Sv* cell (*D*, m) and the distance from the bottom to the *Sv* cell ($D_b$, m). For all models a Gaussian distribution was used for error structure and the link function was the identity function. Additionally, the GCV.Cp method was used for smoothing parameter estimation, and the normality and homogeneity of data were verified.

## Results

### Seasonal variations in fish abundance

Comparing the acoustic data estimations with the set net catches during the survey period, it was found that seasonal changes in fish school density tended to be the same from two data (Table 2). For both years, the fish school density was lowest in autumn, and increased in winter and spring. However, in the acoustic data, the largest fish school density was observed during spring, whereas in the catch data, it was observed during winter. Additionally, based on acoustic data estimations, fish school density was higher in autumn and winter during the LM, compared to that during the NLM.

From the annual total catch, fish catches were much higher (by a factor of approximately three) during the LM than that during the NLM (Table 3). In particular, the catches of Yellowtail and Japanese horse mackerel showed the highest increase out of the total catch, by factors of nine and four, respectively. In addition, the peak catch occurred in February during the NLM and in March during the LM.

### Seasonal variations in fish composition

In summer, fish species were not identified during NLM because the set net and the fishing surveys were not conducted. However, the fishing results during LM show that the catches of bullet tuna (*Auxis rochei*) was the highest, accounting for more than 90% of the total catch (Table 4). In autumn, the set net and fishing catch of Japanese horse mackerel were the highest for both years, accounting for more than half of the total catch. In winter, the set net catch of Japanese horse mackerel was higher in both years. Additionally, the fish species with the highest catch using the set net were Yellowtail during NLM, and Barracuda (*Sphyraena pinguis*) during LM. In spring, larval sardines, including round big-eye sardine (*Etrumeus teres*) and Japanese sardine, and larval Chub mackerel, were most caught both on set net and fishing

**Table 2. Estimated mean surface density and fish abundance using the data of set net catches in each survey (except summer).**

| | Survey | Mean Sa(dB) | Mean density(g/m²) | Estimated biomass in study area(kg) | Catch of Set-net in survey period(kg) |
|---|---|---|---|---|---|
| NLM | Autumn (Dec) Aug. 9–10 | -47.1 | 4.7 | 582 | 237 |
| | Winter (Feb) Dec. 8–10 | -39.7 | 20.6 | 2573 | 4527 |
| | Spring (May) May 9–11 | -35.7 | 46.9 | 5875 | 2538 |
| LM | Autumn (Nov) Aug. 8–10 | -45.5 | 6.4 | 797 | 731 |
| | Winter (Feb) Nov. 25–26 | -39.1 | 25.2 | 3151 | 1779 |
| | Spring (May) May 10–11 | -35.5 | 46.0 | 5752 | 1048 |

**Table 3. Annual catch of each fish species in the set net (Top 10 species).**

| | NLM | | | | | | | | | | | |
| --- | --- | --- | --- | --- | --- | --- | --- | --- | --- | --- | --- | --- |
| Fish species | 201610 (kg) | 201611 (kg) | 201612 (kg) | 201701 (kg) | 201702 (kg) | 201703 (kg) | 201704 (kg) | 201705 (kg) | 201706 (kg) | 201707 (kg) | Total (kg) | Total/year (kg) |
| *Seriola quinqueradiata* | | | 171 | 869 | 6840 | 11873 | 651 | 511 | 289 | | 21203 | 121739 |
| *Trachurus japonicus* | 485 | 2453 | 4983 | 8580 | 8848 | 4055 | 445 | 3497 | 1468 | 1095 | 35908 | |
| *Scomber japonicus* | | 328 | 2025 | 11944 | 12548 | 2256 | 4631 | 1710 | 361 | 140 | 35945 | |
| *Sardinops melanostictus* | | | | | | | 772 | 2582 | | | 3354 | |
| *Seriola dumerili* | 67 | 944 | 1262 | 131 | 66 | 52 | 307 | 151 | 181 | 84 | 3244 | |
| *Coryphaena hippurus* | 73 | 210 | 131 | | | 17 | 1245 | 991 | 97 | 16 | 2781 | |
| *Sphyraena pinguis* | 156 | 1579 | 557 | 452 | 57 | | 39 | 26 | 48 | 31 | 2944 | |
| *Auxis rochei* | 229 | 1555 | 592 | 86 | | | 11 | 31 | 13 | 3424 | 5940 | |
| *Trichiurus lepturus* | 48 | 509 | 597 | 180 | 114 | 160 | 45 | | | | 1653 | |
| *Selar crumenophthalmus* | 329 | 1720 | 3455 | 2521 | 522 | | | | 155 | 65 | 8766 | |
| Total (kg) | 1387 | 9299 | 13772 | 24762 | 28995 | 18413 | 8146 | 9499 | 2611 | 4855 | 121739 | |
| | LM | | | | | | | | | | | |
| Fish species | 201710 (kg) | 201711 (kg) | 201712 (kg) | 201801 (kg) | 201802 (kg) | 201803 (kg) | 201804 (kg) | 201805 (kg) | 201806 (kg) | 201807 (kg) | Total (kg) | Total/year (kg) |
| *Seriola quinqueradiata* | 15 | 152 | 295 | 332 | 1301 | 103421 | 81903 | 1821 | 92 | 135 | 189468 | 339179 |
| *Trachurus japonicus* | 1197 | 1529 | 24138 | 33787 | 6875 | 1051 | 1650 | 436 | 1562 | 2665 | 74889 | |
| *Scomber japonicus* | 67 | 805 | 5754 | 7365 | 860 | 4266 | 2665 | 4015 | 568 | 157 | 26522 | |
| *Sardinops melanostictus* | | | | | | | | | 9854 | | 9854 | |
| *Seriola dumerili* | 1337 | 866 | 2500 | 69 | 170 | | 24 | 1899 | 582.2 | 524 | 7971 | |
| *Coryphaena hippurus* | 559 | 336 | 17 | | | | 722 | 2766 | 962.8 | 36 | 5398 | |
| *Sphyraena pinguis* | 11 | 25 | 63 | 219 | 3412 | 641 | 103 | 47 | 179 | 519 | 5217 | |
| *Auxis rochei* | 687 | 159 | 247 | 316 | | | 11 | 10 | 111.3 | 38 | 1580 | |
| *Trichiurus lepturus* | 81 | 403 | 572 | 12983 | 518 | 272 | 339 | 976 | | | 16144 | |
| *Selar crumenophthalmus* | 101 | 221 | 1700 | 115 | | | | | | | 2137 | |
| Total (kg) | 4054 | 4496 | 35285 | 55186 | 13136 | 109651 | 87418 | 11969 | 13912 | 4073 | 339179 | |

during the NLM. However, during the LM, the set net catch of Yellowtail, and fishing catch of larval sardines and Chub mackerel, were the highest, respectively.

## Seasonal variations in fish distribution

In summer, based on horizontal distribution analysis, fish schools tended to have higher distributions on the shore side than on the offshore side in both years (Figs 3A and 4A). However, the abundance of scatterers on offshore side during LM was decreased compared to NLM (Fig 4A). Vertically, in both NLM or LM, scatterers were most common at a depth of approximately 20 m (Fig 5A), and close to the seabed (Fig 5A'). Additionally, no fish school was detected in the areas shallower than 10 m in both years (Fig 5A). Since the *p*-value of spatial structure exceeded the 5% level in both autumn, it has partial negative significant association (Table 5). Therefore, the reliability of the estimated curve needs to be reconsidered. However, an increase in the number of fish schools during the LM compared with the NLM could be observed from the plot data (Figs 3B and 4B). In particular, the scatterer abundance was increased near the coast and the seabed (Figs 4B and 5B'). In the NLM winter, the scatterer

**Table 4. Results of species composition, weight of fish in set net catches (expect summer), and number of fish by fishing in each survey period (except summer of NLM).**

| | NLM | | | | | | |
| --- | --- | --- | --- | --- | --- | --- | --- |
| | Summer (Aug) | Autumn (Dec) | | Winter (Feb) | | Spring (May) | |
| Fish species | Fishing Number | Set net Weight(kg) | Fishing Number | Set net Weight(kg) | Fishing Number | Set net Weight(kg) | Fishing Number |
| *Seriola quinqueradiata* | - | 28 | - | 3380 | - | 210 | - |
| *Trachurus japonicus* | - | 129 | 7 | 852 | - | 14 | 1 |
| *Scomber japonicus* | - | 70 | - | 295 | - | 399 | 101 |
| *Sardinops melanostictus* | - | - | - | - | - | 1192 | 5 |
| *Seriola dumerili* | - | - | - | - | - | 8 | - |
| *Coryphaena hippurus* | - | - | - | - | - | 534 | - |
| *Sphyraena pinguis* | - | 10 | - | - | - | 3 | - |
| *Auxis rochei* | - | - | - | - | - | 13 | - |
| *Engraulis japonicus* | - | - | - | - | - | | 35 |
| *Etrumeus teres* | - | - | - | - | - | 165 | 27 |
| *Saurida undosquamis* | - | - | 5 | - | 1 | - | 1 |
| | LM | | | | | | |
| | Summer (Aug) | Autumn (Nov) | | Winter (Feb) | | Spring (May) | |
| Fish species | Fishing Number | Set net Weight(kg) | Fishing Number | Set net Weight(kg) | Fishing Number | Set net Weight(kg) | Fishing Number |
| *Seriola quinqueradiata* | - | 30 | - | 33 | - | 910 | - |
| *Trachurus japonicus* | 6 | 672 | 15 | 774 | - | 69 | - |
| *Scomber japonicus* | 2 | 13 | 1 | 51 | 4 | 28 | 92 |
| *Sardinops melanostictus* | - | - | - | - | - | - | 12 |
| *Seriola dumerili* | 5 | - | - | - | - | - | - |
| *Coryphaena hippurus* | - | 13 | - | - | - | 17 | - |
| *Sphyraena pinguis* | - | 3 | - | 921 | - | 21 | - |
| *Auxis rochei* | 53 | - | - | - | - | 3 | - |
| *Engraulis japonicus* | - | - | - | - | 4 | | - |
| *Etrumeus teres* | 3 | - | - | - | - | - | 14 |
| *Saurida undosquamis* | 5 | - | - | - | - | - | - |
| *Seriola quinqueradiata* | 1 | - | - | - | - | - | - |
| *Decapterus maruadsi* | 1 | - | - | - | 3 | | 2 |

The set net data shows only the top 10 types, and fishing data shows only pelagic fish.

abundance was highest near the coast, and decreased with distance from shore. Contrasting, the scatterer abundance has two peaks in the LM winter; the first peak was near the shore, and the second peak came around 2,000 m from the shore (Figs 3C and 4C). In the vertical structure, most of the fish schools were distributed near the seabed in both years; however, during the LM, the distribution of fish schools was dense even near the middle layer of 30 m (Fig 5C and 5C'). In spring, the fish schools were widely distributed over the coastal area to the offshore area (Fig 3D). However, compared to NLM, the scatterer abundance on offshore side was higher during the LM (Fig 4D). Additionally, as well as the fish school was concentrated near the seabed, it was distributed in all depths. The abundance of scatterer was highest in the depth zone up to 30 m for both years (Fig 5D and 5D').

## Seasonal variations in the marine environment

In both years, noticeable stratification occurred in summer, while these layers disappeared in autumn and winter (Fig 6). In summer, water temperature changes significantly from the

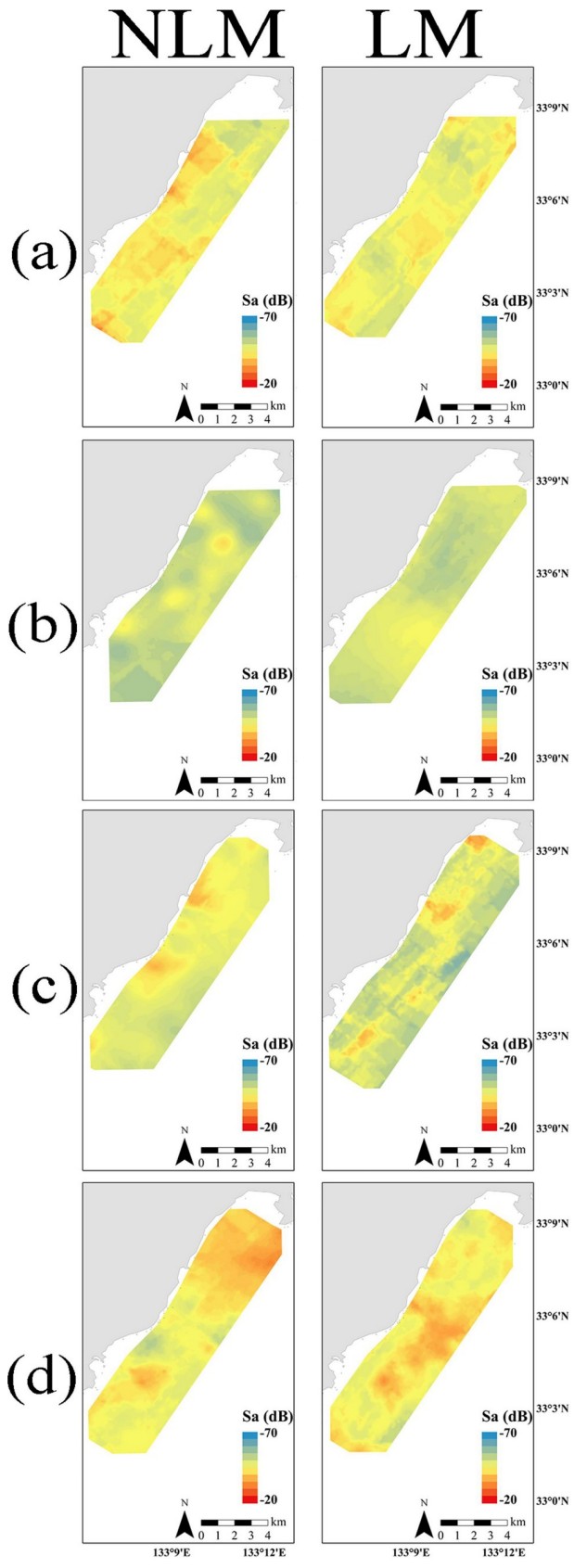

**Fig 3.** Horizontal structure in fish distribution from the ArcGIS Kriging function, expressed as the area backscattering strength $S_a$ and distance from shore $L$ (a: Summer, b: Autumn, c: Winter, d: Spring).

surface to the bottom, 20°C—30°C during the NLM and 19°C—29°C during the LM (Fig 6A). In autumn, almost no difference between the NLM and LM was observed, the water temperature was maintained at approximately 21°C across all depths (Fig 6B). However, water temperatures in winter differed significantly between the two years. The water temperature was 14°C

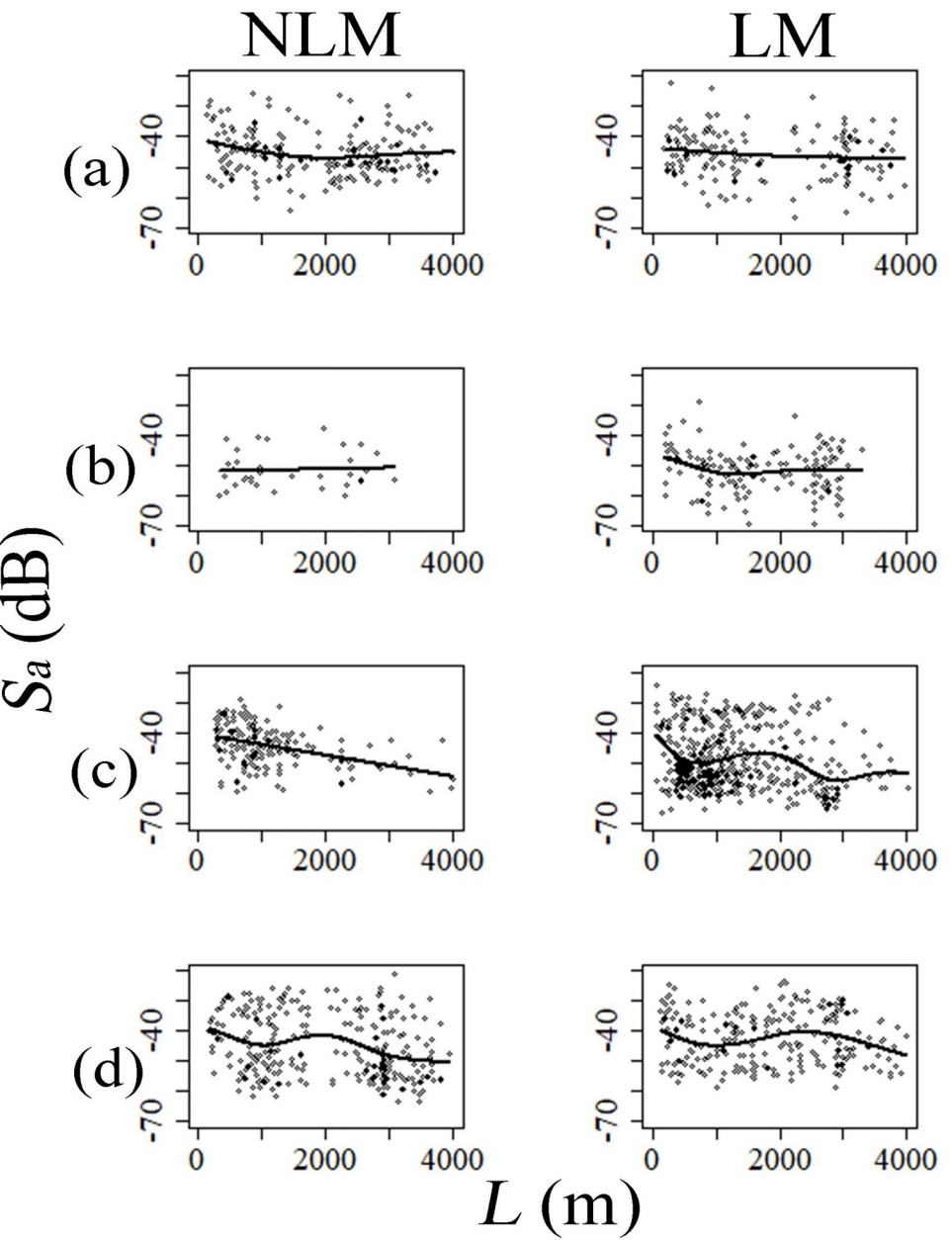

**Fig 4. Seasonal variation of fish distribution in horizontal structure expressed as the relationship between *Sa* (area backscattering strength) and *L* (distance from shore) based on GAM.** The solid line shows the estimated line of fish density (*Sa*). The open circles show the measured value of *Sa* (a: Summer, b: Autumn, c: Winter, d: Spring).

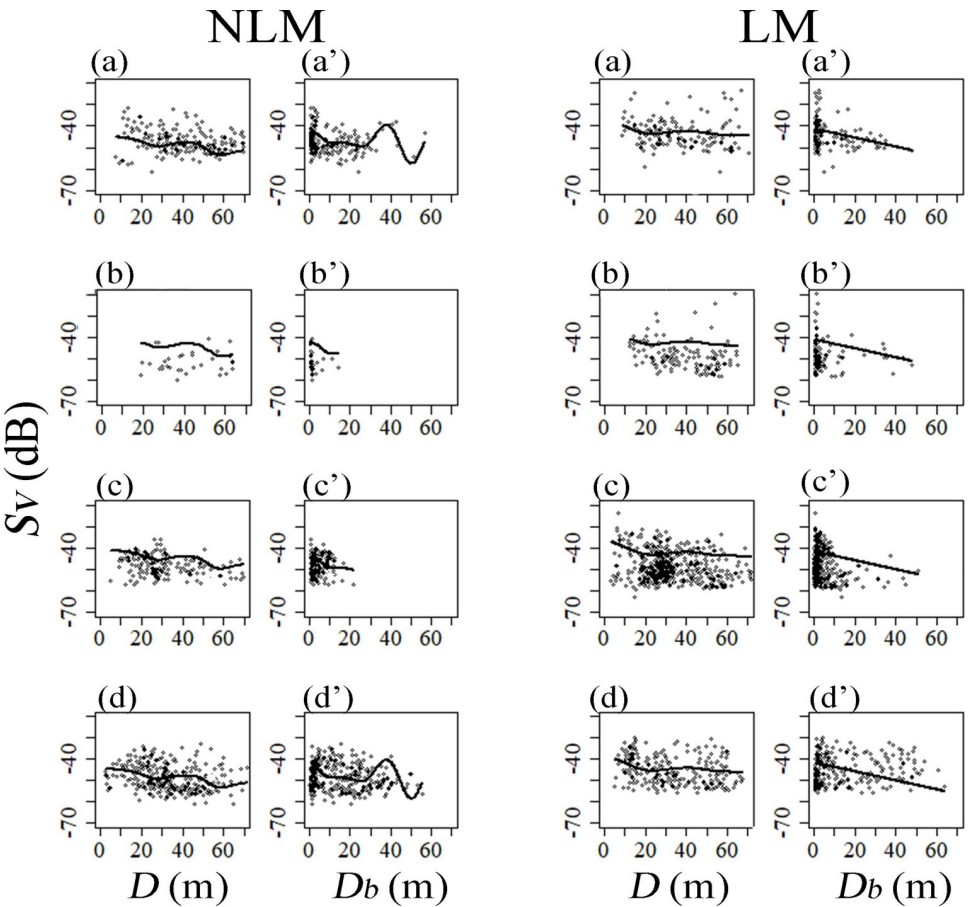

**Fig 5. Seasonal variation of fish distribution in vertical structure expressed as the relationship between *Sv* (volume backscattering strength) and *D* (depth of fish school), *Sv* and *$D_b$* (depth from bottom) based on GAM.** The solid line shows the estimated line of fish density (*Sv*). The open circles show the measured value of *Sv*. The pictures of [a–d] show the data about *D*, and the pictures of [a'–d'] show the data about *$D_b$*, (a and a': Summer, b and b': Autumn, c and c': Winter, d and d': Spring).

—15˚C during the LM, which was lower than that during the NLM by approximately 3˚C (Fig 6C). The difference in water temperature for two years decreased in spring. The minimum and maximum seawater temperatures were 18˚C near the bottom and 21˚C near the surface during the NLM, respectively, and 17˚C and 20˚C during the LM, respectively (Fig 6D).

## Discussion

Comparing the water temperature in survey period during the NLM and during the LM, it tended to be lower during the LM in all seasons. The water temperature during the LM is considerably lower than during the NLM especially in winter. It was also reported that both water and air temperatures remained low during the LM from 2017 to 2018 [34]. Especially in January and February, the mean sea surface temperature of Tosa Bay was the lowest it has been for the last 30 years [34]. Therefore, it is considered that the coastal area in Tosa Bay was widely cooled during the LM. Additionally, since the Kuroshio current is far from the shore during the LM, the habitat of the fish species that ride on the Kuroshio varies greatly, depending on the change of flow path of the Kuroshio [35]. Such changes in sea conditions during the LM are considered to have a significant impact on fish abundance and seasonal dominant fish species.

**Table 5. Statistical results of relationship between *Sa* and *L*, *Sv* and *D*, *D~b~* in each survey.** *L*: distance from shore; *D*: depth; $D_b$: distance from bottom The effective degrees of freedom (edf), which is summary statistic of GAMs, were used to quantify the strength of non-linearity in driver response relationships.

**NLM**

| | Summer (Aug) | | | Autumn (Dec) | | | Winter (Feb) | | | Spring (May) | | |
|---|---|---|---|---|---|---|---|---|---|---|---|---|
| | $S_a$ | $S_v$ | | $S_a$ | $S_v$ | | $S_a$ | $S_v$ | | $S_a$ | $S_v$ | |
| | s(L) | s(D) | s($D_b$) | s(L) | s(D) | s($D_b$) | s(L) | s(D) | s($D_b$) | s(L) | s(D) | s($D_b$) |
| Estimate | -45.6 | -47.2 | | -51.7 | -51.8 | | -44.3 | -48.0 | | -45.5 | -48.4 | |
| edf | 2.42 | 6.10 | 8.29 | 1.00 | 5.84 | 4.33 | 1.00 | 1.00 | 3.16 | 4.49 | 3.97 | 5.39 |
| *p*-value | <0.05 | <0.05 | <0.05 | 0.722 | 0.177 | 0.144 | <0.05 | <0.05 | <0.05 | <0.05 | <0.05 | <0.05 |
| Adjusted $r^2$ | 0.05 | 0.29 | | -0.03 | 0.25 | | 0.15 | 0.11 | | 0.08 | 0.19 | |
| Deviance explained (%) | 5.6 | 34.4 | | 0.4 | 47.9 | | 15.3 | 12.5 | | 9.6 | 21.3 | |
| GCV score | 51.26 | 22.35 | | 40.09 | 25.89 | | 41.23 | 19.30 | | 94.34 | 29.34 | |

**LM**

| | Summer (Aug) | | | Autumn (Nov) | | | Winter (Feb) | | | Spring (May) | | |
|---|---|---|---|---|---|---|---|---|---|---|---|---|
| | $S_a$ | $S_v$ | | $S_a$ | $S_v$ | | $S_a$ | $S_v$ | | $S_a$ | $S_v$ | |
| | s(L) | s(D) | s($D_b$) | s(L) | s(D) | s($D_b$) | s(L) | s(D) | s($D_b$) | s(L) | s(D) | s($D_b$) |
| Estimate | -46 | -43.3 | | -51.8 | -48.9 | | -49.3 | -48.7 | | -42.7 | -45.2 | |
| edf | 1.394 | 3.64 | 1.00 | 2.96 | 3.50 | 1.00 | 6.12 | 6.82 | 6.51 | 4.02 | 2.51 | 7.42 |
| *p*-value | 0.129 | 0.446 | <0.05 | 0.070 | <0.05 | 0.215 | <0.05 | <0.05 | <0.05 | <0.05 | <0.05 | <0.05 |
| Adjusted $r^2$ | 0.017 | 0.09 | | 0.05 | 0.12 | | 0.09 | 0.28 | | 0.05 | 0.17 | |
| Deviance explained (%) | 2.490 | 11.5 | | 6.7 | 14.5 | | 9.7 | 29.9 | | 6.7 | 20.0 | |
| GCV score | 56.538 | 43.08 | | 47.97 | 41.51 | | 86.24 | 33.56 | | 70.62 | 35.24 | |

Based on the annual catch data, the catches of Yellowtails were far higher during the LM than during the NLM. Yellowtail was abundantly caught from February to March during the NLM, and from March to May during the LM. The water temperature suitable for Yellowtails is known to be 16°C—17°C [36]. In winter, the water temperature was 16°C—18°C during the NLM but 15°C or lower during the LM. It is presumed that Yellowtails did not migrate into the survey area in winter during the LM because the seawater temperature was extremely low. However, because the formation of cold-water masses retained the seawater temperature zone suitable for Yellowtails until spring during the LM [34], the Yellowtails migration occurred late, while the Yellowtail fishing season was lengthened. The previous study in Tosa bay showed the same result with our study, the Yellowtail catch might increase during the LM [37]. Therefore, it has been suggested that the catch of Yellowtails tends to increase in the target areas during the LM with low water temperatures, and the migration of Yellowtail depends on the water temperature in Tosa Bay. Japanese horse mackerel catches also increased significantly during the LM, after the Yellowtail catches. The months when Japanese horse mackerel was most caught are January and February during the NLM, December and January during the LM. Since the highest catch of Japanese horse mackerel usually occurs in the water zone at 20°C—21°C [38], the fishing season during the LM with low water temperature was earlier than that during the NLM. In addition, an inner counter-current develops in the Kuroshio Current when it takes the large meandering path [39]. When the inner counter-current is induced, Japanese horse mackerels are supplied into the coastal areas [40]. This physical supply mechanism could result in improved fishing conditions for Japanese horse mackerels. Unlike these two species, since the larval pelagic fishes riding the Kuroshio tend to be moved offshore during the LM [35], the small-sized pelagic fishes in spring has decreased significantly during the LM.

Due to changes in the dominant fish species and abundance in some seasons during the LM, the distribution characteristics of the fish school also changed. Japanese horse mackerel

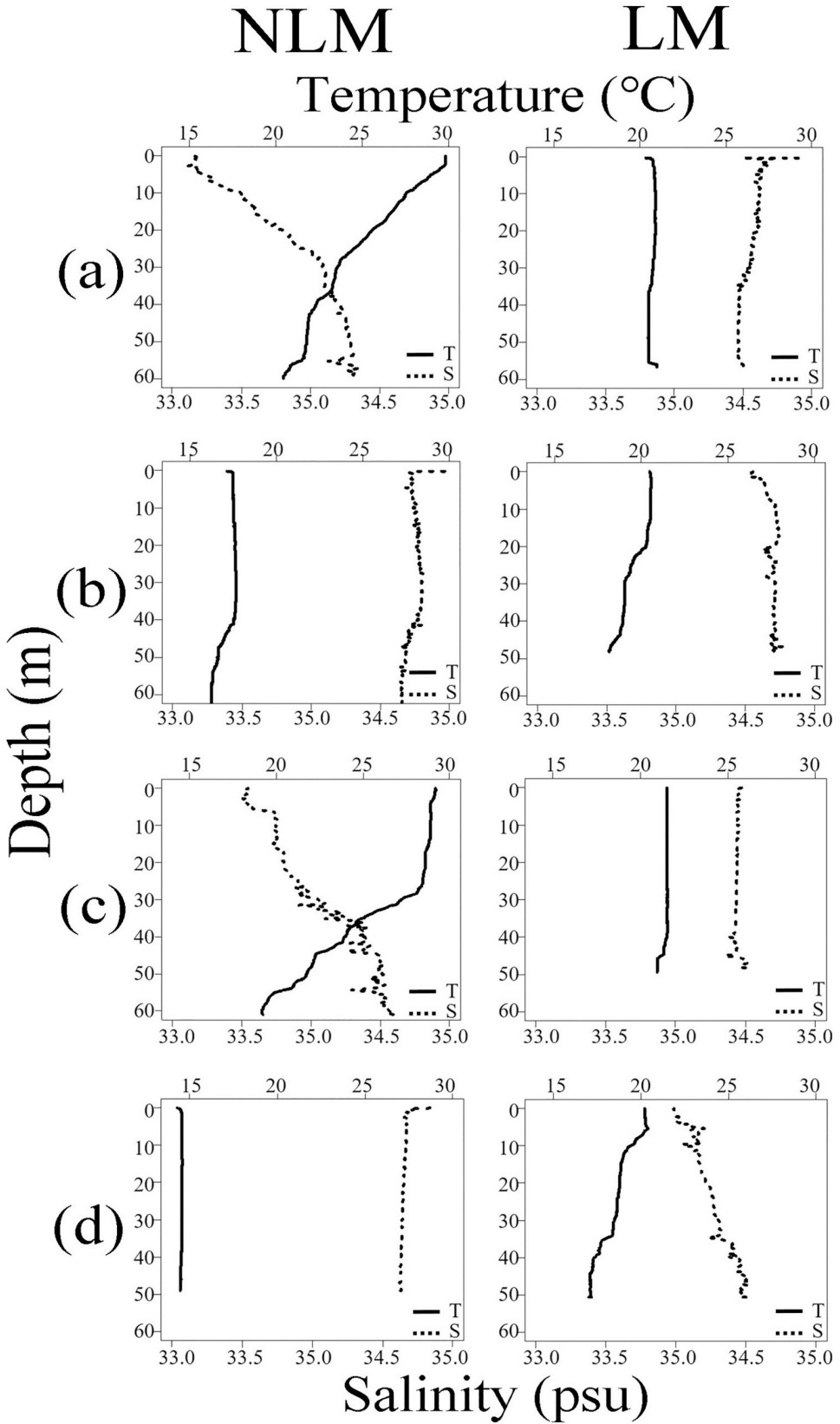

**Fig 6. Seasonal vertical distribution of temperature and salinity around set net.** The solid and the dotted lines indicate temperature and salinity, respectively (a: Summer, b: Autumn, c: Winter, d: Spring).

was the most caught in the fall of both years; however, the LM year experienced higher catches than NLM. Japanese horse mackerels shift to inhabiting the seabed layer and coastal areas when they become adults [41, 42]. Therefore, the scatterer abundance was stronger near the seabed on the shore side during the LM than during the NLM. In winter, many Yellowtails were caught during the NLM, but almost no Yellowtail were caught during the LM while many barracuda were caught. Yellowtails, which migrate into the target areas from the North, are generally distributed in the deep zone at a depth of approximately 50 m [43, 44]. Conversely, barracuda inhabit the sea surface layer in coastal areas [45]. This could explain the increased scatterer abundance in the middle layer in winter during the LM. In spring, many larval sardines and chub mackerels were caught during the NLM, but these floating larval species decreased with the increase in Yellowtails catch during the LM. Larval sardines and chub mackerels have a coastal area distribution [46, 47], but Yellowtails migrate from northern offshore into the Tosa bay for spawning [48]. The different life stages of these dominant fish species could explain the high distribution in the offshore side during the LM.

As described above, there is a difference in abundance and distribution pattern of the fish school that migrate to Tosa Bay under the two Kuroshio Current paths. This difference has been attributed to the fish species and fish behavior affected by the change in coastal sea conditions caused by the LM. In addition, since the effects of the LM differ depending on the fish species, it is necessary to consider LM influences on each fish species rather than collectively.

## Future prospects

In Tosa Bay, fishing season for Yellowtails tends to be longer, and the catches tend to increase during the LM. In the Suzu set net fishery, the installation time of the trap net for Yellowtail is two months. Therefore, it is considered that the long-term installation of Yellowtail trap net is effective in increasing the catch during the LM. In addition, scatterer abundance increased on offshore side in all seasons during the LM. The leader net of the set net was installed 300 m away from the coast only in Suzu, which is closer to the coastal side. Therefore, to increase the catch future, installing the offshore leader net when the Kuroshio current follows the LM pattern is recommended. However, because nets are expensive, effective net improvements need further consideration.

Furthermore, the set net fishery in Suzu was conducted twice daily for a year, except during summer. Since the fish abundance in autumn is small even in the two Kuroshio Current paths, it is possible that the fishing only once daily will be efficient. Additionally, bullet tunas generally migrate into coastal areas along Tosa Bay in summer [49]; however, there were high temperatures in the areas shallower than 10 m and lower fish density. It is presumed that this high water temperature is not a suitable habitat for fish because photosynthetic activity in summer deteriorates due to photo inhibition in surface water [50]. Therefore, placing a moratorium on fishing in summer is the best solution.

As in our study, it has been reported that the catches of Japanese horse mackerels and Yellowtails along the western coast of Sagami Bay and the eastern coast of Izu are higher during the LM than during the NLM [51]. However, it has been reported that catches of sardines and chub mackerels are lower in Kumanonada during the LM [40], which is different from the results of our study. This suggests that the effects of the LM may depend on the survey area. In addition, since the LM can last many years [5], future studies should be conducted over longer time periods for determining any long-term effects on catches in the same study area.

## Supporting information

**S1 Data. *Sa* (area backscattering strength) data extracted from raw data using Echoview.** *Sa* is the decibel value, and *sa* is the linear value. The mean *Sa* value is the decibel value using

averaging the *sa* values.
(XLSX)

**S2 Data. Mean density (g/m$^2$) and whole survey area fish abundance (kg) estimated using mean *Sa* and set net catch ratio.** TS$_{kg}$ is the Target strength (TS) of fish per kilogram. TS$_{kg}$ is the decibel value, and ts$_{kg}$ is the linear value.
(XLSX)

**S3 Data. *L* data calculated with ArcGIS using *Sa* data extracted from raw data.**
(XLSX)

**S4 Data. *Sv* data extracted from raw data, *D* and *Db* data calculated with ArcGIS.**
(XLSX)

**S5 Data. Temperature and Salinity values measured by RINKO-Profiler.**
(XLSX)

## Acknowledgments

We would like to express the deepest appreciation to Marine Fisheries Research and Development Center, Japan Fisheries Research and Education Agency for supporting our study. We would like to thank Masahiko Hamada for navigating the research vessel and the reparation of research cruise. We also thank Editage (www.editage.com) for English language editing.

## Author Contributions

**Data curation:** Yanhui Zhu.

**Formal analysis:** Yanhui Zhu.

**Investigation:** Yanhui Zhu, Kenji Minami, Yuka Iwahara, Kentaro Oda, Koichi Hidaka, Osamu Hoson, Koji Morishita, Sentaro Tsuru.

**Methodology:** Kenji Minami, Hokuto Shirakawa.

**Project administration:** Masahito Hirota, Kazushi Miyashita.

**Writing – original draft:** Yanhui Zhu.

**Writing – review & editing:** Kenji Minami, Yuka Iwahara, Kentaro Oda, Koichi Hidaka, Osamu Hoson, Koji Morishita, Sentaro Tsuru, Hokuto Shirakawa.

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
