## [Decision Letter · Decision Letter 0]

10 Sep 2021

PONE-D-21-18168Seasonal variation in fish school spatial distribution and abundance under the Kuroshio regular pattern and the Kuroshio large meander in Suzu coastal watersPLOS ONE

Dear Dr. Zhu,

Thank you for submitting your manuscript to PLOS ONE. After careful consideration, we feel that it has merit but does not fully meet PLOS ONE’s publication criteria as it currently stands. Therefore, we invite you to submit a revised version of the manuscript that addresses the points raised during the review process.

Please follow up and explain how you have dealt with the reviewer's main concerns. I will evaluate your revised manuscript, maybe based upon a reviewer's opinion. This reviewer will likely be the same as for the original submission so make sure you deal properly with his/her main requirenments.

We look forward to receiving your revised manuscript.

Kind regards,

Geir Ottersen

Academic Editor

PLOS ONE

Journal Requirements:

2. We noted in your submission details that a portion of your manuscript may have been presented or published elsewhere. [The study builds on the work of ‘Seasonal dynamics in fish distribution and abundance revealed by an acoustic survey in coastal waters of the Suzu Area, Kochi Prefecture, Japan’ accepted by Journal of Marine Science and Technology (JMST).The previous paper used one year's data, and this paper uses two years' data.] Please clarify whether this [conference proceeding or publication] was peer-reviewed and formally published. If this work was previously peer-reviewed and published, in the cover letter please provide the reason that this work does not constitute dual publication and should be included in the current manuscript.

“This research received funding from the Marine Fisheries Research and Development Center, Japan Fisheries Research and Education Agency (Empirical Research Project for Marine Fisheries Resource Development: Set-net in Suzu, Kochi prefecture, Fiscal Year 2016-2018). We also thank Masahiko Hamada for navigating the research vessel and the reparation of research cruise”.

“Marine Fisheries Research and Development Center, Japan Fisheries Research and Education Agency (Empirical Research Project for Marine Fisheries Resource Development: Set-net in Suzu, Kochi prefecture, Fiscal Year 2016-2018)”

We recommend that you contact the original copyright holder with the Content Permission Form (http://journals.plos.org/plosone/s/file?id=7c09/content-permission-form.pdf) and the following text: “I request permission for the open-access journal PLOS ONE to publish XXX under the Creative Commons Attribution License (CCAL) CC BY 4.0 (http://creativecommons.org/licenses/by/4.0/). Please be aware that this license allows unrestricted use and distribution, even commercially, by third parties. Please reply and provide explicit written permission to publish XXX under a CC BY license and complete the attached form.”

b. If you are unable to obtain permission from the original copyright holder to publish these figures under the CC BY 4.0 license or if the copyright holder’s requirements are incompatible with the CC BY 4.0 license, please either i) remove the figure or ii) supply a replacement figure that complies with the CC BY 4.0 license. Please check copyright information on all replacement figures and update the figure caption with source information. If applicable, please specify in the figure caption text when a figure is similar but not identical to the original image and is therefore for illustrative purposes only. The following resources for replacing copyrighted map figures may be helpful:

USGS National Map Viewer (public domain): http://viewer.nationalmap.gov/viewer/ The Gateway to Astronaut Photography of Earth (public domain): http://eol.jsc.nasa.gov/sseop/clickmap/ Maps at the CIA (public domain): https://www.cia.gov/library/publications/the-world-factbook/index.html and https://www.cia.gov/library/publications/cia-maps-publications/index.html NASA Earth Observatory (public domain): http://earthobservatory.nasa.gov/ Landsat: http://landsat.visibleearth.nasa.gov/ USGS EROS (Earth Resources Observatory and Science (EROS) Center) (public domain): http://eros.usgs.gov/# Natural Earth (public domain): http://www.naturalearthdata.com/

Reviewers' comments:

Reviewer's Responses to Questions

**Comments to the Author**

1. Is the manuscript technically sound, and do the data support the conclusions?

Reviewer #1: Partly

2. Has the statistical analysis been performed appropriately and rigorously? 

Reviewer #1: No

3. Have the authors made all data underlying the findings in their manuscript fully available?

Reviewer #1: Yes

4. Is the manuscript presented in an intelligible fashion and written in standard English?

Reviewer #1: No

5. Review Comments to the Author

Reviewer #1: General comments.

The Kuroshio large meander is an interesting phenomenon and could affect the transportation of eggs and larvae of fishes, the natural mortality, the recruitment success, then resource abundance and finally fishery, which have attracted boosting attentions in studies on fisheries oceanography. The authors use set-net catch and acoustic survey data to distinguish catch composition and distribution under two patterns of the Kuroshio. However, structures of the Introduction, Results and Discussion should be modified to increase their logicality. The Methods need to be replenished to show the rationality of the analytical framework. The English should be improved to increase the readability. Detailed comments are shown below.

1 Introduction. The Introduction needs to be restructured to improve its logicality. The effects of the Kuroshio Current, as well as its different patterns, on local fishery even ecosystem should be further described to highlight significance of this study. The Kuroshio Current and its two patterns should be illustrated with figures for reader who are not familiar with these current and area. In addition, fishery in the coastal area of Tosa Bay should be described, such as information about main species, etc.

In detail, the first paragraph can introduce the Kuroshio Current and its primary characteristics and importance for local ecosystems; the second paragraph can describe the two patterns of the Kuroshio and the potential effects of these two patterns (It was divided into three patterns in Kaneko et al., 2019, doi:10.1111/fog.12403, I wonder why the authors do not reference the newest study) based on previous literatures; the third paragraph needs to introduce set-net fishery in the Tosa Bay, including the targeting species and their main characteristics; finally the aims of this study should be clearly pointed out.

2 GAM. The authors need to provide more details on model structures, such as setting for the distribution family, linking function, smoothing methods, degree of freedom, etc. Model check process (such as the normality and homogeneity) is imperative and should be displayed to ensure the confidence on results, and the ‘gam.check’ routine is useful. Furthermore, the authors construct three groups of GAM, including one links Sa to L, and two link Sv to D and Db. The reason for such analyses should be clarified as D and Db can used as explanatory variables in one group of GAM. In summary, as the dominant analytical method in this study, model details on GAM should be provided for a better understanding for readers.

3 Results. The current results are divided in two parts - one for regular pattern of the Kuroshio, and another for large meander pattern of the Kuroshio. Overmuch descriptions on fundamental and dispersive results greatly distract attention and deduce interest of readers. As this manuscript aims to compare species composition and distribution between the two patterns of the Kuroshio, I strongly suggest that the author reorganize these results with focus on differences in species abundance, composition, and distribution. The attractive results should be directly displayed with clear language.

4 Discussion. Similar to the Results, the Discussion should also be reorganized. Firstly, repeated descriptions on the Results should be avoided. Secondly, inferences should be strongly supported by your results or previous literatures. Thirdly, brief language will help readers to better understand your points.

5 The English should be polished by native speakers to increase the readability.

Specific comment:

1 Line 53. The authors should cite relevant references to endorse this opinion. In my opinion, I think the Warm Current of Mexico Gulf is the largest warm current.

2 Lines 61-67. A conceptual map of the Kuroshio is needed for readers who are not familiar with this area. Also it will be helpful to show the regular pattern and large meander pattern of the Kuroshio, as well as the cold- and warm-water mass when it is in the large meander pattern.

3 Lines 74-75. How can a bay contribute to variations in the Kuroshio path? Or the authors try to say the variations in Kuroshio path contribute to the physicochemical properties of costal area of the bay. This should be clarified.

4 Lines 77-79. To my knowledge, the fishery of set-net is also largely dependent on the current that drive fishes into the cod-end. Although this paper focuses on the seasonal species composition, this should be declared in the introduction.

5 Line 80. “it” should be “its”.

6 Lines 86-90. Statements of migrating areas of walleye pollock and Japanese pearlsides are redundant for this sentence, as it is highlighting the importance and applications of acoustic survey.

7 Lines 109-110. The latest large meander pattern of the Kuroshio should be described in the Introduction instead of in the Methods.

8 Lines 186-188. The authors should provide details on GAM used in this study, such as explanatory and response variables, setting for the distribution family, linking function, smoothing methods, degree of freedom, etc.

9 Lines 197-198. The authors mention that large meander pattern of the Kuroshio happened in the August 2017, while the low density of fish schools in autumn. Can the large meander pattern last for several months? This should be clarified.

10 Lines 197-206. Repeated descriptions on results should be avoided.

11 Lines 219-220. Table 4 shows that a small number of fishes caught in Autumn, which may not support the dominant role of horse mackerel efficiently.

12 Line 239. I think the citation should be “Table 3”.

13 Line 255. Figs. 2 and 3 should be exchanged.

14 Line 289. Legend of Fig. 4 is confusing.

15 Fig. 8. Details on the measure of environmental factors are not well described, resulting in my doubt on the variations in physicochemical properties. Are these results based on measure of environmental factors for every day in the month or just one or several days in the month?

Although the authors note that environmental factors are different between the two patterns of the Kuroshio, the difference could be caused by different dates of the data instead of the patterns.

16 Lines 408-409. “high fish school responses”. I do not understand this phrase.

6. PLOS authors have the option to publish the peer review history of their article (what does this mean?). If published, this will include your full peer review and any attached files.

Reviewer #1: No

---

## [Author Response · Author response to Decision Letter 0]

28 Oct 2021

Introduction

1. The Introduction needs to be restructured to improve its logicality.

 We have restructured the introduction. First, we introduced the main characteristics of the Kuroshio and why the Kuroshio is important for the local ecosystem. Next, we explained the two patterns of the Kuroshio and why we need to focus on the Kuroshio large meander (LM). Third, we introduced Tosa Bay, which is greatly affected by the Kuroshio Current, and set the net fishing industry in detail. Next, we explained why I had to use the acoustic method. Finally, the purpose of this study was pointed out.

2. It was divided into three patterns of Kuroshio Current in Kaneko et al., 2019.

 The Kuroshio path south of Japan exhibits a remarkable bimodal feature: the large meander (LM) path and the non-large meander (NLM) path. The NLM path is sometimes further classified into two paths, namely, the offshore NLM (oNLM) and the nearshore NLM (nNLM). 

[Usui, 2013 https://link.springer.com/article/10.1007/s10872-013-0197-1] 

[Toida, 2019 https://agriknowledge.affrc.go.jp/RN/2030927543.pdf]

 And, the offshore NLM (oNLM) and the nearshore NLM (nNLM) have the same effect on our study area. [Sugimoto, 2020 https://doi.org/10.1007/s10872-019-00531-8]. In the paper [Kaneko, 2019], the survey area is different from our study area. Therefore, there is only one type of NLM in our study area.

3. A conceptual map of the Kuroshio is needed for readers who are not familiar with this area.

 The Kuroshio paths during the survey period (Fig 1) were drawn using data from Hydrographic and Oceanographic Department Japan Coast Guard. And water masses were modified from previous research. These information was added in the introduction.

4. In my opinion, I think the Warm Current of Mexico Gulf is the largest warm current.

 We changed the sentence to “The Kuroshio Current is one of the warmest water currents in the world”.

5. To my knowledge, the fishery of set-net is also largely dependent on the current that drive fishes into the cod-end. Although this paper focuses on the seasonal species composition, this should be declared in the introduction.

We added the information about seasonal species composition with the change of Kuroshio Current in the introduction. As the sentence “In addition, the fish composition structure in Tosa Bay is seasonal, and the seasonal changes tend to differ depending on the berthing and undocking of the Kuroshio Current that year”.

6. Can the large meander pattern last for several months? This should be clarified.

From the previous researches, the large meander pattern appears once in a roughly ten-year cycle, and continues for more than 1 year. Additionally, the latest large meander occurred in August 2017 is the second longest in history.

[Nishiyama, 1980. https://doi.org/10.2467/mripapers.31.43]

[Miyama, 2019 https://agriknowledge.affrc.go.jp/RN/2010927526.pdf]

 

Methods

7. The authors need to provide more details on model structures.

 The information about distribution family, linking function, smoothing methods, degree of freedom, explanatory and response variables was added in the methods. 

8. The reason for such analyses should be clarified as D and Db can used as explanatory variables in one group of GAM.

 We put the explanatory variables D and Db into one group, and performed a new analysis of the relationship with Sv.

9. Model check process should be displayed to ensure the confidence on results, and the ‘gam.check’ routine is useful.

 We used the ‘gam.check’ to ensure the confidence on results. However, there are four result diagrams for one explanatory variable, and it is quite difficult to include the results of each season for two years for the three explanatory variables in the paper.

 As an example, I posted one results for each response variable here.

 Instead of these diagrams, the results about p-value, adjusted r2, Deviance explained (%) and GCV score were summarized in Table 5.

10. Details on the measure of environmental factors are not well described.

 The marine environment (temperature and salinity) around the set net was measured for one day during each survey period using a RINKO-Profiler. Results of previous research are cited as supplementary information. Our survey results were not different from the monthly average water temperature for the same month.

[Ihara, 2019 https://agriknowledge.affrc.go.jp/RN/2030927527.pdf]

 In addition, the temperature data from Kochi Prefecture Fishing Sea Condition Information System also shows that the difference in environmental factors was caused by different Kuroshio patterns instead of the different dates. 

  

Results and discussions

11. I strongly suggest that the author reorganize these results with focus on differences in species abundance, composition, and distribution.

 We have divided the results in three parts: 1) seasonal variations in fish abundance, 2) seasonal variations in fish composition, 3) seasonal variations in fish distribution, 4) seasonal variations in the marine environment, and focused on these difference between two patterns of Kuroshio Current. 

 For discussions, first, we talked about the marine environment affected by Kuroshio LM. Next, we discussed the fish species composition and abundance that changes due to the influence of the Kuroshio LM. After that, we discussed changes in fish school distribution due to changes in fish composition and abundance.

12. “high fish school responses”. I do not understand this phrase.

 The word for the reflection intensity of fish school has been changed from "response" to "scatterer abundance". [Tournier, 2021 https://doi.org/10.1016/j.jmarsys.2021.103608]

13. The English should be polished by native speakers to increase the readability

 This paper has been edited by editors from ‘Editage’ to ensure language and grammar accuracy. We have submitted an English proofreading certificate to PLOS ONE.

---

## [Editor Report · Decision Letter 1]

8 Nov 2021

PONE-D-21-18168R1Seasonal variation in fish school spatial distribution and abundance under the Kuroshio regular pattern and the large meander in Suzu coastal watersPLOS ONE

Dear Dr. Zhu,

Thank you for submitting your manuscript to PLOS ONE. You have improved your manuscript substantially since the version originally submitted.You have dealt well with the reviewer's requirements and only some minor issues remain. Therefore, we invite you to submit a revised version of the manuscript were the following minor issues are dealt with:The English is now of good standard, but please replace "them" with "it" in line 57.In line 387 please replace a "." with ",". Please order the Figures according to number. Now Figure 1 is last. I don't understand your naming of the Supporting information. You have S2 Table 1, S2 Table 2, S4 Fig, S5 Fig, and S6 Fig. It is very good that you include the excel sheets with data, but why call them figures?Please rename the Supporting information simply as S1, S2, S3, S4, S5, S6 When these small issues are fixed and you have submitted the revised ms I will accept it without further review.

We look forward to receiving your revised manuscript.

Kind regards,

Geir Ottersen

Academic Editor

PLOS ONE
---

## [Author Response · Author response to Decision Letter 1]

11 Nov 2021

1. We have changed "them" with "it" in line 57, and replace a "." with "," in line 386. 

2. We ordered the Figures according to number.

3. We renamed the Supporting information simply.

---

## [Editor Report · Decision Letter 2]

15 Nov 2021

Seasonal variation in fish school spatial distribution and abundance under the Kuroshio regular pattern and the large meander in Suzu coastal waters

PONE-D-21-18168R2

Dear Dr. Zhu,

Thank you for following up on my last few instructions. We’re pleased to inform you that your manuscript has been judged scientifically suitable for publication and will be formally accepted for publication once it meets all outstanding technical requirements.

Kind regards,

Geir Ottersen

Academic Editor

PLOS ONE

---

## [Editor Report · Acceptance letter]

17 Nov 2021

PONE-D-21-18168R2 

Seasonal variation in fish school spatial distribution and abundance under the Kuroshio regular pattern and the large meander in Suzu coastal waters 

Dear Dr. Zhu:

I'm pleased to inform you that your manuscript has been deemed suitable for publication in PLOS ONE. Congratulations! Your manuscript is now with our production department. 

Kind regards, 

on behalf of

Dr. Geir Ottersen 

Academic Editor

PLOS ONE